# Technological Tools for the Early Detection of Bovine Respiratory Disease in Farms

**DOI:** 10.3390/ani12192623

**Published:** 2022-09-30

**Authors:** Andrea Puig, Miguel Ruiz, Marta Bassols, Lorenzo Fraile, Ramon Armengol

**Affiliations:** 1Department of Animal Science, ETSEA, University of Lleida, 25198 Lleida, Spain; 2Agrotecnio Research Center, ETSEA, University of Lleida, 25198 Lleida, Spain

**Keywords:** advanced technologies, behavior, bovine respiratory disease, cattle, early diagnosis

## Abstract

**Simple Summary:**

The inclusion of remote automatic systems that use continuous learning technology are of great interest in precision livestock cattle farming, since the average size of farms is increasing while time for individual observation is decreasing. Bovine respiratory disease is a main concern in both fattening and heifer rearing farms due to its impact on antibiotic use, loss of performance, mortality, and animal welfare. Much scientific literature has been published regarding technologies for continuous learning and monitoring of cattle’s behavior and accurate correlation with health status, including early detection of bovine respiratory disease. This review summarizes the up-to-date technologies for early diagnosis of bovine respiratory disease and discusses their advantages and disadvantages under practical conditions.

**Abstract:**

Classically, the diagnosis of respiratory disease in cattle has been based on observation of clinical signs and the behavior of the animals, but this technique can be subjective, time-consuming and labor intensive. It also requires proper training of staff and lacks sensitivity (Se) and specificity (Sp). Furthermore, respiratory disease is diagnosed too late, when the animal already has severe lesions. A total of 104 papers were included in this review. The use of new advanced technologies that allow early diagnosis of diseases using real-time data analysis may be the future of cattle farms. These technologies allow continuous, remote, and objective assessment of animal behavior and diagnosis of bovine respiratory disease with improved Se and Sp. The most commonly used behavioral variables are eating behavior and physical activity. Diagnosis of bovine respiratory disease may experience a significant change with the help of big data combined with machine learning, and may even integrate metabolomics as disease markers. Advanced technologies should not be a substitute for practitioners, farmers or technicians, but could help achieve a much more accurate and earlier diagnosis of respiratory disease and, therefore, reduce the use of antibiotics, increase animal welfare and sustainability of livestock farms. This review aims to familiarize practitioners and farmers with the advantages and disadvantages of the advanced technological diagnostic tools for bovine respiratory disease and introduce recent clinical applications.

## 1. Introduction

The global demand and production of livestock products is increasing rapidly due to population growth, rising incomes and changes in lifestyles and diets [1]. The global demand for meat is expected to increase by at least 40% in the next 15 years, and beef is one of the most important sources of animal protein [2]. This might become a problem, since the number of farmers is decreasing every year. As a result, the size of farms and the number of animals per farm is increasing, making the proper clinical diagnosis of diseases a big challenge [2].

Bovine respiratory disease (BRD) is one of the most studied cattle diseases since the end of the 19th century [3] and its incidence has increased in recent years [4], constituting a great challenge for veterinarians and producers, since it represents the greatest economic loss due to mortality and morbidity within the young stock rearing and feedlot sectors [5,6]. These economic losses occur because of a decrease in daily weight gain and yield, an increased conversion rate and treatment costs. Therefore, a lower carcass quality, increased mortality and lower animal welfare is commonly occurring [7,8,9]. Its prevention and diagnosis can be difficult, because BRD is a complex and multifactorial disease that involves the interaction of several factors: environment, infectious agents and the host [3,7]. Most cases of BRD in cattle feedlots occur in the first three weeks after arrival [5,7] and account for 75–80% of disease cases during this period [10]. BRD occurs more frequently during fall and winter [11]. Risk factors include stressful situations, such as transportation, handling (loading, unloading and habituation), dehydration or hunger due to lack of water or feed, or not adapting to the type of drinkers or feeders on the new farm, mixture and regrouping of animals of different origin, overcrowding, extreme temperatures, high levels of dust, high levels of humidity and poor ventilation [7,12]. BRD is usually caused by a virus (mainly bovine respiratory syncytial virus (BRSV), parainfluenza virus type 3 (PI3), bovine coronavirus (BCoV), bovine herpesvirus type 1 (BoHV-1) and bovine viral diarrhea virus (BVDV)) and bacteria (mainly *Mannheimia haemolytica, Pasteurella multocida, Histophilus somni* and *Mycoplasma bovis*) [3,7,13]. Most of these bacteria can be found as commensals in the upper respiratory tract [14]. Viruses normally cause a first infection that immunosuppresses and damages the respiratory tract and the animals’ defense mechanisms, and secondarily, bacteria proliferate and colonize the lung, causing pneumonia. Animals that have an increased risk of suffering BRD include the lowest body weight calves within a group, the youngest individuals, calves that are sick due to other causes, immunocompromised individuals, animals with poor nutritional status and calves that have not been vaccinated or have had no prior exposure to pathogens [3,7].

Sick animals suffer a significant decrease in meat quality, welfare, health, productive efficiency and an increase in production costs per produced animal [1,15]. BRD is the second most frequent pathology in pre-weaned calves, immediately after diarrhea. However, it is the main cause of morbidity and mortality in weaned heifers and fattening units [16,17,18,19]. Therefore, BRD is the main concern in heifer rearing and cattle feedlots worldwide [5,8,20]. Diagnosis of BRD in cattle is usually based on visual assessment of clinical signs, with low sensitivity (Se) and specificity (Sp) [21]. To improve disease detection, technological tools have been developed that objectively measure health and welfare automatically [22,23]. These new technological tools are included under the term precision livestock farming (PLF), defined as “the management of individual animals through continuous real-time monitoring of health, welfare, production/reproduction and environmental impact” [2,24]. The objective of PLF is not to replace humans, but to support them [25]. Furthermore, these tools allow the management of a large number of animals and an early diagnosis of diseases [26]. Using this approach, veterinarians have a list of animals suspected of having BRD that is generated by the system and sent to their phones, tablet or computer, that helps them in making diagnostic and treatment decisions [23,27].

The setup of this list of “animals suspected to be sick” is possible due to the data received by the technological devices used on the animals when carrying out continuous monitoring of their behavior. Data are transmitted to software that stores and analyzes the information. When the system detects an alert (using mathematical algorithms), it provides the information to the farmers‘/practitioners’ device. The system is a dynamic and continuous learning process based on algorithms that are continuously updated by the contributions of the farmer/practitioner [25]. The more data that are compiled, the more the system learns and, therefore, a more accurate BRD diagnosis may be achieved (Figure 1). The accuracy of the BRD diagnosis and treatment through PLF may be increased when laboratory diagnosis is integrated. Thus, metabolomics may be able to differentiate BRD-infected from vaccinated animals [28] and predict disease outcomes after secondary bacterial infection [29]. Moreover, antimicrobial susceptibility testing could also be integrated into big data analysis to improve treatment for BRD.

The early diagnosis of pathologies is one of the primary objectives in current livestock production [24], which can be achieved thanks to constant monitoring and individualized data from the animals. Treating sick calves at an early stage, before the onset of clinical signs, reduces the use of antibiotics (ATB) and improves their effectiveness, a good strategy to deal with antimicrobial resistance (AMR) [30]. The increase in AMR in bovine respiratory bacterial pathogens poses a threat to BRD control [31]. This strategy fits well with regulations on veterinary medicinal products (Regulation (EU) 2019/6 of the European Parliament and of the Council, of 11 December 2018) that are a cornerstone for achieving the objectives established in the “From Farm to Fork” strategy and the “One Health” European Action Plan to fight AMR [32]. On the other hand, by improving the effectiveness of treatments with the new technological tools, the animals would have milder clinical signs, a shorter duration of pathology and a higher cure rate, so there would be an improvement in health outcomes and animal welfare [33]. This would lead to greater social acceptance of intensive production systems, increasingly questioned by society [34]. Finally, better animal welfare is also positively correlated with improved performance and better sustainability of livestock farms [35]. With these new technological tools, it is possible to increase productive performance [2], since the cure rate increases compared with conventional clinical diagnostic techniques that detect subclinical animals with this system [7]. Such technology could prevent BRD clinical cases from becoming chronic, with permanent and generalized pulmonary fibrosis, adhesions or abscesses, which have a negative impact on the productive performance of the animal and increase the risk of recurrence or relapse and mortality [7,9].

The main aim of this work is to review new technological tools for early diagnosis of BRD for young stock rearing and cattle feedlots. To tackle the main objective, three specific sub-objectives have been developed: (1) To demonstrate the need for an early diagnosis method for BRD. For this, the disease and the main current diagnostic methods are described, indicating their shortcomings. (2) Review the existing technology for the early detection of BRD. (3) Describe the strengths, weaknesses, opportunities and threats of these technological tools.

## 2. Materials and Methods

This study is a bibliographic review. Four main databases were used in the literature review: Google Scholar, PubMed, Scopus and Web of Science (accessed from November 2021 to July 2022), and a total of 24 keywords were used (Table 1). To identify the relevant studies within the coincidences, a reading of the abstracts and conclusions of all the manuscripts and proceedings was carried out by the first author, eliminating those that did not deal with the main topic. Finally, websites were used mainly for the addition of images with the aim of making it easier for the reader to understand the content.

## 3. Results and Discussion

A total of 105 papers were included in this review, with 73 studies on beef cattle and 25 on dairy. In total, 13 papers were studies about activity behavior, 15 were about feeding behavior and 11 papers were about spatial behavior. The rest of the references included have been used to introduce key topics and discuss some of the relevant technological issues.

### 3.1. Conventional BRD Diagnostic Methods

Diagnostic methods for BRD that require sampling can be on in vivo or post mortem biological samples. In vivo biological samples are based on blood samples, nasal or nasopharyngeal swabs and bronchial and/or transtracheal lavage, using techniques such as PCR, ELISA and microbiological culture. Post mortem sampling is based on tissue samples subjected to laboratory techniques such as histopathology, microbiological culture and PCR [34]. On the other hand, there are diagnostic methods that do not require sampling, such as conventional clinical diagnosis. This can be complemented by other diagnostic methods, such as auscultation [36], clinical scoring [37,38], diagnostic imaging [39,40], post mortem lung observation [41] and even the evaluation of the response to treatment [42].

It is important to highlight that clinical diagnosis is currently the most widely used method to identify cattle sick with BRD. In addition to clinical signs, this diagnosis is also based on behavior changes. Behavioral changes and clinical signs in BRD-sick cattle include ocular and nasal discharge, cough, extended head, dry snout, floppy ears, poor coat, dull eyes, lethargy, social isolation, hypo-/anorexia, pyrexia, dehydration, tachypnea and dyspnea [7,27]. Clinical diagnosis may not be accurate and does not allow early detection of BRD, because the clinical and pathological presentations of this syndrome are varied [5,8,21,43]. It has been estimated that this diagnostic method has a low Se and Sp, of 62 and 63%, respectively [21].

The evaluation of behavior by practitioners, technicians and farmers to diagnose cases of BRD under field conditions is a challenge because other bovine diseases or environmental conditions, such as metabolic acidosis, lameness or heat stress, can cause behavioral patterns similar to BRD [44]. Furthermore, cattle display prey behavior in attempting to disguise the clinical signs of their disease during a clinical exam. Due to the lack of manpower, observations of behavior are often short, subjective and may not reach consistent conclusions [21,27,45,46].

Clinical scoring systems have been developed in recent years to improve early and accurate detection of calves affected by BRD. These mechanisms use a simple and objective clinical scoring system, standardizing the identification of sick calves and increasing the Se of clinical diagnosis without increasing costs [37,38]. These scoring systems are useful as they assign values to clinical signs, giving each animal a total score, which corresponds to the risk or probability of becoming clinically ill [47], but they do not differentiate between upper or lower respiratory tract disease and do not identify calves with subclinical pneumonia [48]. Note that they are designed for specific age groups; for example, for pre-weaned [38] or weaned calves [34,37].

To increase the Se and Sp in the diagnosis of BRD, visual evaluation can be combined with diagnostic imaging, which is a non-invasive method that allows the diagnosis of pneumonia in vivo. The main drawback of radiography is that it cannot be used in a farm, due to limitations in equipment, expense, anesthesia requirements and potential radiation exposure [48]. However, ultrasound is a non-invasive diagnostic tool that is increasingly used to evaluate the lungs and pleura, as a source of complementary information to clinical examination [40]. Thoracic ultrasound (TUS) can be performed on farms using portable devices [48]. The weakness of diagnostic imaging techniques is that they detect BRD when the animals already have lesions. This is exactly what technological tools aim to solve.

### 3.2. Automated Registration Systems

Automated registration systems monitor cattle continuously with no need for human presence. Therefore, this approach may increase the likelihood of identifying subtle changes in early or mild stages of the disease that could be masked by the calf or not detectable by visual assessment [45,49]. Early diagnosis of diseases with the use of sensors reduces the use of ATB [30], improves animal welfare [33] and reduces relapse, recurrence and mortality rates [50]. Since 2014, the number of published studies about the early detection of BRD has increased with the monitoring systems focused on early detection behavior. Farmers and veterinarians should not see behavior assessment systems (BAS) as a threat, but as a tool that makes their work easier and accurate [27]. These systems are “automatic tools capable of continuously recording the activity, behavior, physiology and weight of animals” [51].

BAS is based on individual identification of the animals. Thus, animal electronic identification (EID) has become essential in the development of PLF. This EID establishes a unique number, and the information is transferred from the transponder to the transceiver (that is, from the EID animals’ device to a reader) using a radiofrequency identification system (RFID) [52,53]. The most often used devices for EID are ear tags, pedometers, collars, ruminal boluses and subcutaneous microchips [35,54].

Most of the technological tools that are detailed in this review use this RFID system. An RFID system is mainly composed of three elements (Figure 2): an RFID tag or transponder, an RFID reader or transceiver and a computer with specific software. The transponder can be embedded in an object (i.e., pedometer) or in an animal (i.e., electronic ear tags). It does not use batteries, so it is considered passive technology. The tag includes a microchip that stores data and an RFID antenna that sends the information recorded to the RFID reader or transceiver. The RFID reader or transceiver sends the signal to the RFID tag, reads the information and then sends this information to the software [35,54].

RFID can work with three different levels of radio frequency: low frequency (LF, 125–134 kHz), high frequency (HF, 13.56 MHz) and ultra-high frequency (UHF, 860 MHz) [35]. If UHF is used, the data are collected in real time; on the other hand, with LF or HF, the data are downloaded at regular intervals [2].

#### 3.2.1. Behavior Monitoring Tools

Recognized behavioral patterns in animals at the onset of febrile infectious diseases are lethargy, depression, hypo- or anorexia and reduced water intake. These patterns are expected in sick animals compared to healthy ones [24,27].

Currently, there are three main technological options to assess the behavior of cattle for early BRD detection: (1) Monitoring of physical behavior using three-axis accelerometers. These quantify the number of steps animals walk, the time they are standing/lying down and the number of times they lie down [9,27,55,56]. (2) Systems that monitor behavior at the feeder and drinker through sensors [56,57]. These quantify daily feed/milk/water intake, the time animals are eating/drinking and the number of times they eat/drink [8,57,58,59]. (3) Monitoring of spatial behavior using global positioning systems (GPS) or real-time location systems (RTLS), to continuously determine the location and movement of calves within the pen [27,56]. All technological tools use the RFID system to transmit information, with the exception of GPS, which uses wireless fidelity (Wi-Fi).

Continuous and non-invasive monitoring of cattle behavior using various technologies has the potential to be a powerful health management tool in fattening [8] and dairy farms [57,60,61], and may allow earlier detection of diseases compared to conventional clinical assessment methods [55,62,63].

##### Physical Behavior Monitoring

Accelerometers have been widely used in dairy production systems for estrus detection and locomotion problems; however, their use in beef cattle is less explored [61]. These devices have gained acceptance in cattle research because they allow a greater understanding of an animal’s movement and behavior on an ongoing basis. Accelerometers can be used as pedometers, ear tags or collars [56]. Loss of accelerometers is more common for pedometers than ear tags, as commercial pedometers are designed for dairy cows and do not fit well on grazing calves because they are smaller [27]. In addition, the placement of pedometers on the animal can temporarily alter the normal behavior of the calves and temporarily affect movements, until the animal becomes habituated [22].

Accelerometers continuously measure the gravitational force in multiple axes, and the values are processed to determine the activity and postural behaviors of the animals [22]. Accelerometers have been shown to accurately monitor standing, lying or walking behaviors of calves with 97.7% agreement with video analysis, considered the gold standard for evaluating other behavioral monitoring methods [22,64,65]. Accelerometers can also monitor rumination [66]. It has been widely demonstrated that monitoring rumination can be very useful for accurately predicting and diagnosing multiple disorders in dairy cows [67]. Although it is not widely used in fattening units, it has been demonstrated that animals suffering from BRD had a lower rumination index compared to healthy ones at three to six days before the onset of clinical signs [68].

Most of the accelerometers that are currently used in livestock farms consist of detection on three axes. In addition, they have the ability to continuously record the data and summarize at intervals set by the user (every 15, 30, 60 or 120 min) and may have remote sensing capabilities using UHF radio waves that allow real-time analysis. All three axes are recorded simultaneously and, depending on the placement of device on the animal, the XYZ plane differs between the position of the upright calf and the position of the calf lying down, allowing both positions to be distinguished [56]. The *Y* axis is perpendicular to the ground when the animal is lying down, while the *X* axis is perpendicular to the ground when the calf is standing [64].

On-farm implementation of this system is simple. It requires an accelerometer containing both sensors that detect movement and the standing/lying down position, as well as a transponder, made up of a microchip that continuously collects data from the animal and an antenna that occasionally sends the information to the transceiver (reader). Information regarding physical behavior will arrive in the computer software along with the transponder number and the animal’s identification number (Figure 3).

The decrease in behavioral variables begins between four and six days before the clinical diagnosis of BRD, and is more accentuated the day before the clinical identification of the disease [55]. Pillen et al. [55] determined that an animal was clinically ill by assessing its state of illness and depression with a clinical score. They determined the number of steps taken by calves from an accelerometer placed near the metatarsus of the right hindlimb, and found a significant difference between calves with a subsequent clinical diagnosis of BRD and calves that did not become ill in the number of steps since day 4 before the clinical diagnosis of BRD. This confirms the results of White et al. [69], in a study in which cattle were inoculated with *Mycoplasma bovis*. These authors found that the number of steps was negatively associated with a clinical disease score and the size of lung consolidation, suggesting that step monitoring reliably detected disease and differentiated severity. The same results were observed after *Mannheimia haemolytica* inoculation—the number of steps decreased compared to the control group [70]. On the other hand, Tomczak et al. [71] evaluated the average daily minutes of activity recorded with an accelerometer and found differences between the control group and the BRD cases. It should be noted that the differences in activity between calves treated for BRD and those not treated for BRD existed only between 8:00 and 20:00 h of the day. For this reason, the study concluded that for the early detection of BRD, it is only necessary to monitor the behavior of the animals during this period, reducing the amount of data needed and making it easier to implement new behavior assessment systems [27,71]. In contrast to this, in other studies, two-step peaks were reported, one from 5:00 to 11:00, and the other from 16:30 to 21:00 in the controls and until 19:30 in the test animals [55]. In the last peak of activity, there was a greater difference in steps between BRD cases and controls. Feed supply took place between 6:00 and 7:30, and between 12:00 and 12:30. This would indicate that the bimodal pattern may have been due to the type of feed distribution. However, Smith et al. [33] observed a similar bimodal pattern in daily activity counts in calves with a single feed delivery (7:00 to 9:00).

A decrease in activity due to BRD could be confused with lameness or locomotion problems [56], although the visual evaluation of the calves by the farmer or veterinarian should differentiate between the two. Reduced activity in sick animals is a biologically protective mechanism that involves decreasing metabolic costs to save energy for the immune system and the indirect effects of febrile and inflammatory responses to infection. [71,72]. In 2013, Theurer et al. [70] determined that calves experimentally inoculated with *Mannheimia haemolytica* increased resting time compared to the control group on the day the inoculation was performed. The same findings were observed in another study, where calves spent more time lying down after exposure to an experimental challenge [73]. In 2016, Pillen et al. [55] determined the mean time spent standing and active, and found a significant difference between calves with a subsequent clinical diagnosis of BRD and calves that subsequently did not become ill in the time spent standing and active one day before clinical identification of the illness. However, the values in the standing and active time observed in the BRD cases were disparate. Individual animal environmental conditions and circadian rhythms also affect the time calves spend lying down, so it is important to consider these factors when comparisons are carried out [70,74].

These findings suggest that other variables evaluated (i.e., number of steps during the day) may be more effective than the time spent standing and lying down to diagnose early BRD, since it follows an irregular pattern [22]. It should not be forgotten that cattle with severe BRD prefer to stand for a long time to facilitate breathing, while milder cases remain recumbent for long periods to save energy [72].

In summary, it could be stated that all the variables of physical behavior monitored with accelerometers are significant and useful for an early BRD diagnosis, except for the number of times calves lie down. Activity provided by the accelerometers can be used as an objective method to help in the management and early identification of sick calves with BRD. If the cost–benefit and the sensors’ attachment to calves are improved, this technology could be effective and feasible for the early detection of BRD in calves.

##### Behavior at the Feeder and Drinker

Several systems have been developed to detect BRD-sick animals by monitoring eating behavior and individual feed intake in group-housed cattle [22,56]. The data needed to monitor eating behavior are the number of visits to the feeder/drinker during the day and their time lapse, as well as the amount of feed/water ingested. These variables can be determined using RFID technology with UHF systems, either with ear tag microchips or transponders placed on collars [75,76]. For its set up in the farm, a system that accurately measures the amount of feed or water consumed by the animals is needed, and this is normally done with feeding stations. Measurement is possible using integrated sensors that weigh the amount of feed and/or water ingested. During the pre-weaning period, the automated milk feeders (AMF) provide accurate information such as milk consumption, drinking speed and the number of rewarded and unrewarded visits to the feeder, which could potentially be used to predict disease development [57]. Monitoring individual time spent at the feeder and the frequency of daily visits requires RFID individual tags that also capture the EID of the animal. RFID antennas should also be strategically placed near the drinkers and feeders to detect the presence of a specific EID. This system has a peculiar characteristic, since the transponder is separate; the animal carries the microchip, but the antenna is part of the feeding station (Figure 4) [77]. The information regarding the intake of water and feed, and time spent at the feeder and drinker will be compiled with the RFID reader and will be transmitted wirelessly to software able to report data on the health, performance and intake of each animal [56].

Such systems require compartmentalization of the feeding area. At arrival to the facility, animals can be frightened and generally require time to habituate. This can be a problem mainly when new grazing calves arrive in the feedlot, during which time they have a high risk of BRD. If access to the feeder/drinker is not easy, this stress may exacerbate this risk, triggering BRD cases. In addition, system errors or confounders may occur until the calves become accustomed to it, since in the first days their consumption would decrease in a way uncorrelated with health status. To address this issue, other systems use antennas enabled above an open feed line, allowing the determination of the frequency and duration of eating. However, data on the average individual feed and water intake are not available [56].

Monitoring eating behavior has been shown to be a promising tool for the detection of BRD and predicting morbidity and mortality in cattle [8,22,57]. A study [79] monitored behavior at the feeder in recently arrived fattening calves and observed that during the first four to five days, the calves decreased the frequency and duration of staying at the feeder. On the other hand, [59] also found that BRD-sick calves made fewer visits to the feeder compared to healthy calves on the day they were diagnosed with BRD by feedlot technicians.

Studies quantifying changes in feeding behavior relative to the time of the visual detection of BRD are scarce but could provide enough information for the commercial development of automated disease detection systems. By monitoring feeding behavior with RFID, it is possible to diagnose BRD-sick calves 4.1 days earlier compared to conventional methods (subjective visual evaluation by farmers with 8–15 years of experience), with a Se of 90% and a positive predictive value (PPV) of 91% [58]. Other studies have been able to predict BRD risk in cattle seven days prior to visual detection, when a calf suffering from BRD was defined as having a rectal temperature ≥40 °C, at least two clinical signs (reluctance to move, dry nose, nasal or ocular discharge, drooping ears or head and poor appearance) and a serum concentration of haptoglobin >0.15 mg/mL. The decrease in average feed consumption, a decrease in the average time spent eating and a decrease in the number of times they were going to eat per day (intake frequency) were significant and could be used as indicators [8,80]. This is because cattle with fever reduce feed intake [72]. The hypo-/anorexia observed in sick cattle is part of a coordinated strategy to combat the disease. Based on this, the animals would rest more and decrease eating to save energy [45]. In addition, as studied in humans, it has been determined that appetite decreases with increased immune response [81].

Several studies have been carried out using AMF behavior monitoring during the pre-weaning period to predict morbidity and mortality caused by BRD. All of these results were recently reviewed, and it was found that daily milk consumption, drinking speed, and rewarded and unrewarded visits may provide insight into early disease detection in pre-weaned dairy calves [57].

On the other hand, the results regarding drinking behavior do not provide conclusive information, so this would not be a good variable for the early detection of BRD [34]. In 1997, [82] established that calves reduced their time spent drinking water by 23.7% between three and four days before showing clinical signs of BRD and, therefore, drinking behavior could be used in the early detection of sick animals. On the contrary, [56,79] concluded that sick calves had a higher frequency and duration at the drinker compared to healthy ones. However, the study by [71] found no significant differences in the time spent consuming water between healthy and sick calves.

##### Spatial Behavior Monitoring

Spatial behavior monitoring can provide additional information on the specific behaviors of cattle. This monitoring in the pen is carried out using RTLS or GPS [56]. GPS can provide an approximate location of animals, but the readings are not precise enough to delineate specific activities, such as going to eat or drink. On the other hand, RTLS are designed to locate the position of an object anywhere within a defined area. Furthermore, RTLS systems have considerably longer battery life than current GPS technology [22,77].

Although the RTLS system is similar to RFID feeding behavior systems, the RTLS has the advantage of being able to monitor an animal’s location anywhere within the pen, so it does not restrict assessment to only feeding and drinking behaviors [56]. The RTLS system consists of receivers placed around the desired tracking space, RFID tags (transponders) that are placed on the animals to be monitored and software able to receive and process location data [22]. Additionally, RTLS works through communication between an EID tag and the readers that control the area. At least three readers must communicate with the EID tag at any point within the coverage area; this is known as triangulation [56]. RTLS has been used for different purposes: (1) To quantify the changes that follow painful procedures and the changes associated with the state of well-being [56]. (2) To determine location information to quantify the true patterns of contact between animals, which can be useful for explaining the transmission of diseases within the pen [83,84]. Since previous work has shown that cattle contact patterns are dynamic within the pen and vary between calves, depending on the time of day and on the day [85,86], a better understanding of true contact patterns can be helpful when designing prevention or intervention techniques for communicable diseases within a population [56]. (3) To quantify the time spent at the water trough and the distance traveled, concluding that the distance traveled can be associated with the level of lung consolidation [69].

These systems have been shown to be an effective method for identifying BRD with higher Se and Sp compared to visual observations [84,86,87]. It was also found that pre-weaned calves experimentally inoculated with *Mannheimia haemolytica* reduced grooming and social interactions for two days after inoculation, suggesting that these behavioral changes may be good indicators of early stages of respiratory disease [88]. Interestingly, although sick calves initiated fewer social grooming interactions (i.e., grooming their pen mates), they tended to receive longer episodes of social grooming, suggesting that healthy calves did not avoid sick calves, but quite the contrary. This could lead to infectious diseases being transmitted even more easily between animals. This had been studied previously in rodents and the same conclusion was reached [88].

Remote early disease identification (REDI), as a specific system for early diagnosis of BRD [83], includes as variables in the algorithms the levels of activity, the location within the pen (percentage of time spent near water and feed) and social patterns (i.e., time spent in a group or isolated). Every day, the system creates a report on the health status of the animals, providing an alert if any animal needs clinical assessment and/or treatment. The REDI is able to identify BRD-sick calves up to 0.75 days earlier than a trained operator [85]. The drawback of the REDI system is that it only detects BRD cases, so operators must still check all the pens to detect other health problems such as lameness, digestive and neurological disorders [88]. The diagnosis of BRD with the conventional method (visual observation of clinical signs) and REDI were compared [84], determining that the REDI system has a Se of 81.3% and a Sp of 92.9%, while the visual system has a Se of 64.5% and Sp of 69.1%. The advantages of REDI compared to conventional methods were a reduction in ATB use and an earlier diagnosis of BRD. Differences in morbidity, mortality, performance (ADG) or efficacy in the first treatment were not observed between the two groups (conventional versus REDI system). It is important to highlight that calves of the conventional method group were treated with an ATB preventively, whereas the calves of the REDI system group were not treated [89]. This preventive treatment could explain why differences in performance were not observed. Other authors found that the number of times calves are treated is negatively associated with ADG [90].

The REDI systems not only maintained productive performance, but also increased performance in calves that were treated twice, since the ADG was better in the animals that were under therapeutic treatment and not submitted to ATB prophylaxis. It would be interesting to evaluate the real differences in a study in which neither of the two groups was preventively treated with an ATB [89]. It is important to highlight that [89] did not observe significant differences between morbidity, efficacy of the first treatment and mortality between the conventional method versus the REDI system. This should be considered a success, since other studies indicate that prophylaxis with ATB reduces the expected morbidity by up to 50% [91]. This was possible because the REDI system could accurately identify BRD cases early in the disease process and was probably able to prevent disease outbreaks in the pen. The decrease in ATB use in the REDI group is due to the fact that calves did not receive ATB upon arrival. By obtaining an accurate diagnosis of the disease with the REDI system, it was only necessary to treat calves with BRD, which may be an important strategy to reduce AMR. The reduction of ATB use offers several benefits: lower health costs, lower AMR and better public perception.

### 3.3. Other Tools Available Based on Physiological Changes

Other technological tools based on physiology, especially body temperature, are under study, but they need further development and some of them fail to diagnose BRD at early stages [22,92]. According to Wisnieski et al. [19], automated infrared thermography (IRT) measurement systems detect BRD in calves when they already show clinical signs. These systems measure the temperature of the nostrils and the surface of the nasal plane [22] and use ear tags sensitive to tympanic temperature [92]. It should be noted that IRT has only been able to diagnose early BRD when the measurement was of radiated orbital temperature [93,94], the duration of rumination and rumen temperature with reticulo-ruminal boluses [49,95]. It was concluded that IRT with orbital temperature measurement could be useful in the diagnosis of BRD, as it could diagnose it even up to a week before the onset of clinical signs [94]. In 2012, these same authors automated the non-invasive use of this technique in feedlots through a measurement station located in the drinker [94]. On the other hand, two studies determined with reticulo-ruminal boluses that HR episodes preceded clinical signs of BRD by 12 to 136 h [49] and 10 to 194 h [96]. Biologically, this makes sense, because fever is an adaptive mechanism to combat viral and bacterial infections [72]. A few hours or a few days after infection, the typical peak of fever occurs, which usually goes unnoticed by the feedlot staff [14].

### 3.4. Comparison between Approaches

The weaknesses, threats, strengths, and opportunities (SWOT) of the BAS technological tools for early diagnosis of BRD are detailed in Table 2. On the one hand, they are continuous, objective, individual, non-invasive and remote monitoring systems [23,27]. Thus, they manage to monitor animals’ health without interrupting their natural behavior (except for pedometers and feeding stations that require habituation) [27,45,46]. In addition, being objective means that they do not depend on weather conditions, emotions or worker’s experience [23,27].

Although the Se and Sp of all early diagnosis tools are not known, the REDI systems specifics for BRD do not differ much from those for thoracic ultrasound techniques (Se: 81.3%; Sp: 92.9%) [84]. However, its potential benefits lie in the fact that making an early diagnosis improves the effectiveness rate of ATB and reduces their use [30]. Therefore, this is a way to deal with AMR and, in turn, to improve animal welfare, since early treatment reduces the number of sick animals by reducing transmission and the time of illness per animal [33,34]. As a result, these systems could shorten the healing period and minimize lung lesions that hinder productivity. A full, fast and complete healing would make cattle more resistant to relapse or recurrence of the disease. Furthermore, an early diagnosis of BRD allows an increase in productive performance, because it detects calves with BRD before they have significant lesions and includes the diagnosis of subclinical animals [7].

The main opportunity presented by the BAS technological tools for early diagnosis of BRD, which still need to be developed, is the combination with other conventional BRD diagnostic techniques. An example could be laboratory diagnosis. If the causal agents responsible for BRD clinical signs were periodically detected on a farm and integrated into the software, correlations of microorganisms and behavior patterns unnoticed by humans could be triggered by mathematical algorithms. In the case of bacteria, if antimicrobial susceptibility testing were performed by qualitative (Kirby–Bauer method) or quantitative methods (minimal inhibitory concentration (MIC)), it could guide the practitioner in each “suspected” case of the disease to use the ATB that have frequently been more sensitive to the specific pattern of disease. This could work if all the laboratory diagnostic results were linked to the software. Thus, the software could look for repeated patterns of BRD and ATB, which have been more sensitive to each pattern (in the case of bacteria) (Figure 5). This does not guarantee the success of the approach, but it would be a fast way to carry out the most accurate ATB treatments at the farm level, where not much information may be available. It is clear that a practical way to achieve antimicrobial stewardship in cattle is needed, as has been proposed for other livestock species [97]. As an example, when a particular pattern of disease behavior is detected (i.e., “Pattern A”), the computer program would be able to associate it with a specific pathogen (i.e., “Pathogen A”). With the antimicrobial susceptibility testing previously performed on that farm and correlated by the software with “Pattern A”, reporting to the practitioner the best therapeutic option (i.e., the use of a particular ATB, Option A), in addition to a second and even third ATB treatment option (Options B and C).

This system would require periodic analysis; therefore, the laboratory would be a key player. The practitioner veterinarian would also indicate to the program whether or not the treatment has been effective in each case, so that the system was constantly learning and improving. As a final result, it could improve the cure rate even further and decrease the AMR.

Another promising combination would be the use of BAS with pathogens or biomarker detection. Metabolomics has been used as a DIVA approach to discriminate between BRD-infected and vaccinated animals [28], for classifying animals as BRD or non-BRD [98] and even for predicting the outcome of BRD cases [29]. Results could be integrated into a big data system and, through AI, the software could learn and integrate behavior differences between sick and healthy animals or vaccinated and infected animals, or could provide information about the severity of outcomes. To achieve such combinations, the sampling and metabolite analysis should be continuous, as it is used, for example, in humans with diabetes type 1 [99], in dairy cows for the control of reproductive status through progesterone levels in milk [100,101,102] or monitoring betahydroxybuirate levels in milk for ketosis diagnosis [103,104]. The problem of applying this strategy for accurate and early detection of BRD is the sampling procedure. As an example, if sampling and analytical in situ method could detect increased levels of lactate prior to viral infection and increased glucose levels following viral infection, then practitioners and farmers would have an accurate predictive value of sick cattle survival [29]. To the authors’ knowledge, there is no other commercial type of biological sample other than milk through milking that could be used to continuously monitor metabolites or pathogens in cattle. This is, today, the limiting factor of this combination of BAS with the analysis of biomarkers or pathogens indicative of BRD and its different levels of severity. If a system could be implemented, for example, in the ear tag, to analyze biomarkers and/or the presence of a certain pathogen responsible for BRD (i.e., BHV-1 gE glycoprotein), this would be a great advance [105]. A possible solution could be to integrate the analytical results from some BRD-sick and healthy animals, at some stages of an outbreak, and let the machine learn and correlate the results with the behavior of the animals.

However, these technological tools also have their weaknesses. Perhaps the most important drawback to their implementation is the cost, as they require a significant investment, and the benefits must come in savings from ATB treatment costs, improved performance and reduced mortality rates [27]. Further investigation is needed to analyze the cost–benefit of such investment [22,27]. Another challenge would be the management of the continuous flow of data and the design of mathematical algorithms that achieve a high Se and Sp in BRD diagnosis. In addition, the system must be properly implemented for health management, and a list of “animals suspected to be sick”, which should be clinically explored in the pen, would be routinely generated together with animals that suffer from other illness or cases that would not be detected by the software, since Se and Sp are not 100%. [27]. A practical solution would be to include a signaling mechanism to speed up detection (i.e., a light on the device of the “suspicious” animal or a spray that marks them as they eat or drink).

The absence of pathognomonic behavior associated with BRD makes it difficult to assess both conventional and modern evaluation of cattle behavior, since the clinical signs of BRD can be confused with acidosis, dehydration or heat stress [27,44]. In addition, the calves’ behavior seems to be affected by the environment, particular areas or regions and season [22]. As detailed in this review, there are different methods of early detection of BRD based on BAS, which detects the disease a variable number of days in advance (Table 3). The possibility that the behavior of sick calves might be different according to the predominant clinical signs in each individual case, and even according to which tissues or areas of the respiratory tract are most injured by different pathogens, should also be considered.

Other remarkable threats could be the replacement of staff by robots, although that is currently unthinkable since these technologies demand highly skilled labor for maintenance and repair. These systems may result in a technological dependency, which can lead to impacts related to a lack of supply of system components or a need for electricity or internet connection on the farm [23].

Finally, we also sought to answer whether technological tools represent the future of BRD diagnosis for cattle farms. No conclusive information was found on this topic. Note that while these technologies can make an early diagnosis of BRD, the on-farm implementation of the systems needs to be improved and needs to be more accessible to farmers. In this sense, the setup of these technologies would be easier and more viable for big herds. In any case, it seems clear that the most promising aspect of this type of technology is the use of a single device capable of facilitating two main tasks: (1) monitoring different behavioral variables and integrating them into algorithms [57,65] and (2) providing on-time sampling and analysis of pathogens or biomarkers indicative of BRD [105].

Other factors that can influence the use of such technology include trends in animal welfare, RAM concerns or farming policies that would stimulate quicker uptake of PLF systems in the near future [35]. Moreover, global demand for animal products is increasing, but the number of farms is decreasing, making it more difficult to diagnose diseases such as BRD on larger farms [2]. In countries where a limit on herd size is mandatory, one may assume that the implementation of this technology will not be necessary. However, if the number of farmers decreases and the demand for meat increases, it is believed that farmers will have more animals to care for and will need even more of these technological tools to increase efficiency in rearing a larger number of calves [26].

## 4. Conclusions

BRD is of primary concern in young stock rearing worldwide, as it represents one of the largest economic losses due to mortality and morbidity. Conventional methods for BRD diagnosis have shortcomings, such as subjectivity, difficult detection of subclinical cases and delay (lag) in diagnosis, and require time for training and labor.

Livestock behavior is an important component in the early diagnosis of BRD, and new technological tools can greatly improve the work of farmers and practitioners.

The most commonly used behavioral variables for early detection of BRD are feeding/drinking and activity (number of steps). Other interesting variables such as level of activity, rumination, the location within the pen and social patterns can also be integrated.

Implementing BAS on farms must have a positive cost–benefit ratio, improve efficiency and health parameters of the herd, be reliable and be integrated into existing management practices. Different farm designs and sizes may limit the use of such systems.

The use of single sensors capable of integrating BAS for multiple variables and on-site biological sampling and analysis for biomarkers and/or pathogens seems to be the most promising path for the future of these technologies.

Further research is needed to adapt such systems to farms and to use them as a powerful tool for early detection of BRD.

## Figures and Tables

**Figure 1 animals-12-02623-f001:**
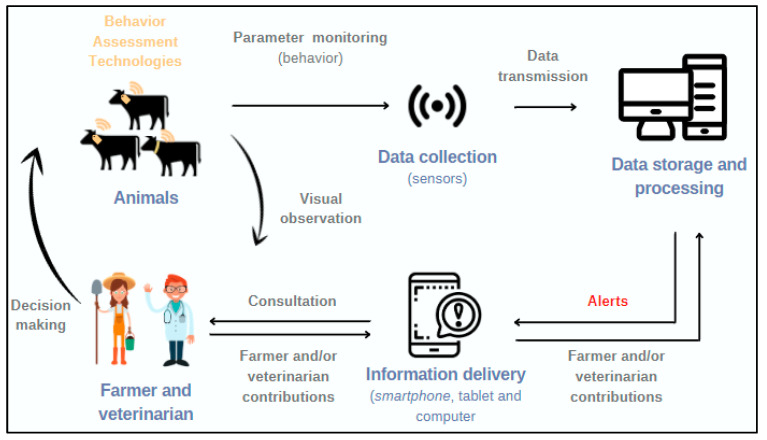
Diagram of health management at a cattle farm when implementing behavior assessment systems (BAS). Adapted from [25] but originally created by the authors.

**Figure 2 animals-12-02623-f002:**
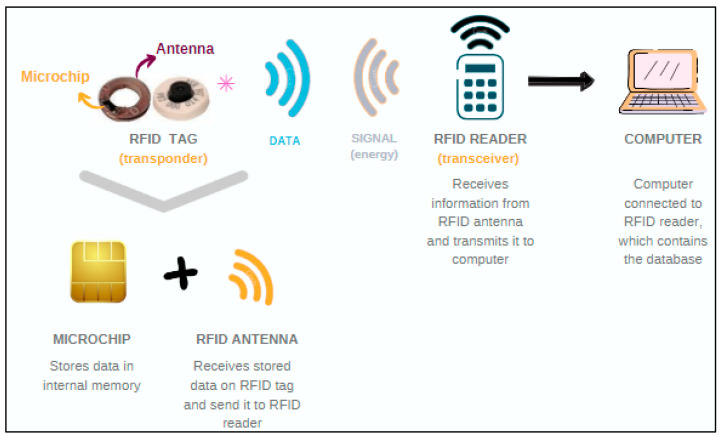
Basic electronic identification (EID) using radiofrequency identification system (RFID) transmission. Note that the RFID tag can have different presentations. This figure shows an example of an electronic ear tag. * Could also be a collar or a pedometer as a transponder. Own source.

**Figure 3 animals-12-02623-f003:**
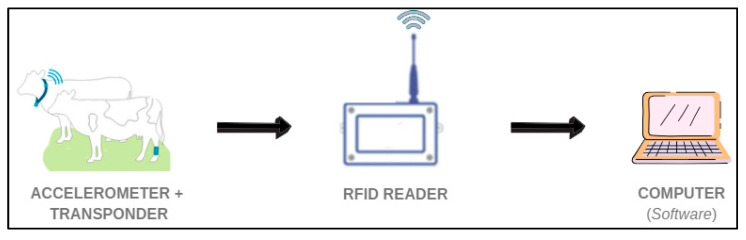
Basic accelerometer system with radiofrequency identification (RFID). Own source.

**Figure 4 animals-12-02623-f004:**
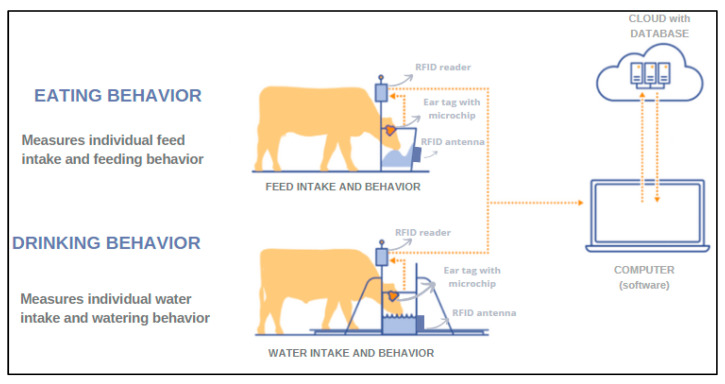
The system individually identifies the animals each time they go to the feeder or drinker and monitors the time they spent there using the radiofrequency identification system (RFID) ear tag. In addition, the system accurately measures the amount of feed/water consumed by that animal through sensors. Data are then transmitted to software. Adapted from [78] but originally created by the authors.

**Figure 5 animals-12-02623-f005:**
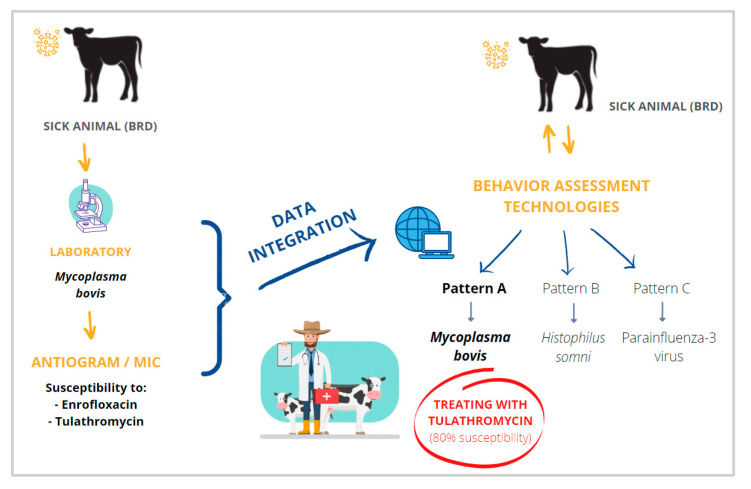
Example of the combination of a conventional diagnosis method (laboratory analysis) with remote early disease identification (REDI) specific for bovine respiratory disease (BRD). Own source.

**Table 1 animals-12-02623-t001:** Details of the search and keywords used in Google Scholar, PubMed, Scopus and Web of Science (accessed from November 2021 to July 2022). For important topics, full words and their most commonly used abbreviations in the scientific literature were used.

Category	Group	Key Words
**General**	BRD ^3^ Precision farming technology tools	“Artificial intelligence” OR “AI ^1^” “Bovine respiratory disease” “Machine learning” OR “ML ^2^” “Metabolomics” “Nanotechnologies” “Precision livestock farming” “PLF ^2^” & “Animal welfare” “PLF ^2^” & “Antibiotics” “BRD ^3^” & “Artificial intelligence” OR “AI ^1^” “BRD ^3^” & “Behavior” “BRD ^3^” & “Diagnosis” “BRD ^3^” & “Early diagnosis” “BRD ^3^” & “Machine learning” OR “ML ^2^” “BRD ^3^” & “Metabolomics” “BRD ^3^” & “Nanotechnologies” “BRD ^3^” & “Precision livestock farming” “BRD ^3^” & “Sickness behavior” “BRD ^3^” & “Technology”“PLF ^4^” & “Trend”
**Specific**	Activity behavior	“BRD ^3^” & “Accelerometer” “BRD ^3^” & “Activity” “BRD ^3^” & “Podometer”
	Feeding behavior	“BRD ^3^” & “Feeding Behavior”
	Spatial behavior	BRD ^3^ & REDI ^5^

^1^ AI, artificial intelligence. ^2^ ML, machine learning. ^3^ BRD, bovine respiratory disease. ^4^ PLF, precision livestock farming. ^5^ REDI, remote early disease identification.

**Table 2 animals-12-02623-t002:** Weaknesses, threats, strengths and opportunities (SWOT) analysis of the use of behavior assessment systems (BAS) as technological tools to diagnose BRD in cattle.

**STRENGTHS**	**WEAKNESSES**
– Continuous, objective, individual, non-invasive and remote monitoring systems – Reduced use of ATB ^2^ – Improved animal welfare – Improved production performance – Make up for the lack of staff – Continuous evolution of accuracy	– Cost (high financial investment) – Improvement of mathematical algorithms – Alteration of the behavior of the animals until they adapt, in the case of pedometers and feeding stations – Daily human check of the pens in search of other pathologies – Affected by housing design of the farms – Lack of pathognomonic changes of BRD ^3^ behavior (i.e., acidosis, heat stress or lameness) – Not all approaches are equally efficient and there is a great amount of variability between them
**OPPORTUNITIES**	**THREATS**
– Reduction of AMR ^1^ – Improved sustainability – Better consumer perception – Technification of farms – Data management – Combination with other diagnostic techniques – Reduction in labor cost	– Replacement of staff by robots if technology improves greatly in the future – Technological dependency (supply, electricity and connection to the network and/or internet)

^1^ AMR: antimicrobial resistance. ^2^ ATB: antibiotic. ^3^ BRD: bovine respiratory disease.

**Table 3 animals-12-02623-t003:** Summary of the days prior to the clinical diagnosis of BRD cases using technology compared to the clinical diagnosis based on BRD clinical signs.

*Technology*	*Clinical Signs Used for Clinical BRD ^1^ Case Definition*	*Days Ahead of the Clinical Diagnosis of BRD ^1^*	*Reference*
Pedometry	Gaunt, nasal/ocular discharge, lags behind other animals in the group, cough, labored breathing, non-responsive to human approach and depression.	4 d.	Pillen et al. [55]
Time standing		1 d.	Pillen et al. [55]
Behavior in the feeding area	Nasal/ocular discharge, cough, depression and inappetence.	4.1 d.	Quimby et al. [58]
	Reluctance to move, crusted nose, nasal/ocular discharge, drooped ears or head, gaunt appearance and rectal temperature (≥39.5 °C).	7 d.	Wolfger et al. [8,62]
REDI ^2^	Depression, lack of appetite, increased respiratory rate, and increased nasal discharge.	0.75 d.	White et al. [86]

^1^ BRD, bovine respiratory disease. ^2^ REDI, Remote early disease identification.

## Data Availability

Not applicable.

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
