# Peer review of "Technological Tools for the Early Detection of Bovine Respiratory Disease in Farms"

_animals, 2022, doi:10.3390/ani12192623_

Round 1
Reviewer 1 Report (Previous Reviewer 2)
Dear authors,
Thank you for this very interesting piece of work which I found meritorious of being considered further for publication.
This being said, I would suggest some hints for its improvement, if authors agree.
I personally think that the core of your paper covers an aspect of absolute importance both for the production (including animal health and welfare as well as sustainability of farming systems) and technological development towards which the scientific community is called to lead the revolution of practices.
The paper is well structured though especially in the introduction some passages gluing the different concepts lack a bit of adequate shifts in detailed writing style, to say so.
In addition, some general concepts are referred only to one or two references which are not exactly depicting the progress of the state of the art, more than once neglected also further in text. Indeed, despite some references were added to this version, I feel that some aspects are still undervalued. Please, see the review by Morrone et al. 2022 on Sensors doi: 10.3390/s22124319 . Additionay see Cappai et al., 2019 on PeerJ Computer Science doi: 10.7717/peerj-cs179
Table 2 it is opportunities and not opportunities, so please add a 'p'
On the whole, as said, the paper is well structured but still needs some additional language polishing for english style, rather than for grammar.
Title is concise and pertinent to the content of the paper. Keywords are well conceived. The paper is balanced in all its parts and well developed (only some writing style polishing). Discussion can be improved with references and deepening of concepts from results. Conclusion is good.
Thank you.
Author Response
Dear Reviewer,
Thank you very much for your comments/corrections and suggestions.
Please see the attachmennt.

Reviewer 2 Report (New Reviewer)
the review article is describing the different advanced technological tools that can be used in the diagnosis of respiratory diseases in cattle.
the review is well organized and well written. the authors put a lot of effort to gain different information about the currently used artificial intelligent tools in cattle farms that can help the owners and practitioners to early diagnose respiratory diseases.
Author Response
Dear Reviewer,
Thank you very much for your comments/corrections and suggestions.
Please see the attachmennt.

Reviewer 3 Report (New Reviewer)
Comments to the authors
L20: .. time-consuming..
L87: .. phones …
L119: … shorter duration of pathology..
L137: .. weaknesses, opportunities, and threats..
L173: .. post-mortem..
L202: … anaesthesia requirements..
L207: ..what another..
L242: add references in figure 2
L281-282:… behavior of the calves …
L297: .. placement of device..
L310: add references in figure 3
L406: … for the detection…
L410: … visits to feeder ..
L426: … hypoxia/anorexia…
L495: .. an earlier diagnosis…
L497: … highlight that calves …
L500: .. differences in performance..
L504: .. was under therapeutic..
L542: … strengths, and opportunities…
L560: .. specifics for BRD…
L578: .. case of the disease…
L599: add references in figure 5
L616: …., if sampling and analytical….
Author Response
Dear Reviewer,
Thank you very much for your comments/corrections and suggestions.
Please see the attachmennt.

Reviewer 4 Report (New Reviewer)
Bovine Respiratory Disease (BRD) is one of the most serious cattle diseases, since it induces the greatest economic loss due to mortality and morbidity within the young stock rearing and feedlot sectors, and the early diagnosis of pathologies is of important in current live stock production. The main aim of this manuscript is to review new technological tools for early diagnosis of BRD for young stock rearing and cattle feedlots. But this manuscript has major limitation as follow:
1. BRD is a complex and multifactorial disease that involves the interaction of several factors: environment, infectious agents, and the host. The clinical diagnosis is currently the most widely used method to identify cattle sick with BRD. But, the manuscripts review the existing monitoring tools for the early diagnosis of BRD. As the abnormal behavior changes of standing, lying, eating, or walking also occur in other disease, it is very hard to achieve the specificity of BRD diagnostic by behavior monitoring tools.
2. In line 406-407, the manuscript described that monitoring eating behavior has been shown to be a promising tool for detection of BRD and predicting morbidity and mortality in cattle. In fact, many diseases in cattle could also cause the low appetite, how could eating behavior be a promising tool for detection of BRD?
3. In line 437-438, how to understand that “drinking behavior do not provide conclusive information, so this would not be a good variable for the early detection of BRD”?
4. In line 476-477, many diseases could cause to reduce grooming and social interactions of pre-weaned calves, such as diarrhea. How did these behavioral changes may be good indicators of early stages of respiratory disease?
5. In line 527, why does IRT has only been able to diagnose early BRD? IRT is just a tool for monitoring body temperature, and there are so many etiologies for the change of the body temperature.
6. Extensive editing of English language and style are required. such as line 51、319、324、336、343 et al.
7. In line 159-160, 104 papers were included in this review, with 73 studies on beef cattle and 25 on dairy, what about other papers?
Author Response
Dear Reviewer,
Thank you very much for your comments/corrections and suggestions.
Please see the attachmennt.

Round 2
Reviewer 4 Report (New Reviewer)
I approve the manuscript for publication
This manuscript is a resubmission of an earlier submission. The following is a list of the peer review reports and author responses from that submission.
Round 1
Reviewer 1 Report
The manuscript is a relatively good summary of the current state of technology as it applies to BRD. However, I was disappointed that the literature search did not include the key works - artificial intelligence (AI), nanotechnologies, and perhaps metabolomics. The technology discussed in the manuscript has been around for many years, but is just now being used in some large North American feedlots. One of the evolving and exciting new areas is the use of artificial intelligence, particularly as it applies to monitoring behavioural data. The manuscript needs to make some reference to 'big data' and the use of AI in analyzing these very large, data-rich databases.
The other aspect that is missing is the use of ear tags embedded with 'chips' to analyze for metabolites that are indicative of early BRD - acute phase proteins, insulin levels, etc. These technologies are already well-known in human medicine and it is only a matte of time before they are applied to livestock production. I think a review of what is happening in the dairy industry - which is arguably much more technologically advanced than the beef cattle industry - would help inform on potential new technologies that could be brought to bare against BRD. Related to this is the new and emerging field of nanotechnologies.
Unfortunately, the manuscript suffers from far too many grammatical and sentence structure issues, which makes reading difficult at times. I also think that it is repetitive in places. It is also inappropriate to use a reference as a proper noun in a sentence - see line 226-227. It should be "According to Vaintrub et al ...." Also, line 332-333, is this correct that the sick calves stand more than the healthy calves, because the next sentence implies the opposite?
Reviewer 2 Report
Dear authors,
I read your review and found it good, however I feel that the literature is already plenty of reviews that cover those aspects. Moreover, aspects you stressed on throughout the text are being debated since years and no new perspectives, in my opinion, come out of your paper, though I found it good in structure, writing and composition.